# *ACADL* Promotes the Differentiation of Goat Intramuscular Adipocytes

**DOI:** 10.3390/ani13020281

**Published:** 2023-01-12

**Authors:** An Li, Yanyan Li, Youli Wang, Yong Wang, Xin Li, Wuqie Qubi, Yan Xiong, Jiangjiang Zhu, Wei Liu, Yaqiu Lin

**Affiliations:** 1Key Laboratory of Qinghai-Tibetan Plateau Animal Genetic Resource Reservation and Utilization, Ministry of Education, Southwest Minzu University, Chengdu 610041, China; la98614@163.com (A.L.); liyanyan@swun.edu.cn (Y.L.); wangylwy@163.com (Y.W.); wangyong010101@swun.cn (Y.W.); lixin97628@163.com (X.L.); qibiwuqie@163.com (W.Q.); xiongyan0910@126.com (Y.X.); zhujiang4656@hotmail.com (J.Z.); liuw@swun.edu.cn (W.L.); 2Key Laboratory of Sichuan Province for Qinghai-Tibetan Plateau Animal Genetic Resource Reservation and Exploitation, Southwest Minzu University, Chengdu 610041, China; 3College of Animal Science and Veterinary Medicine, Southwest Minzu University, Chengdu 610041, China

**Keywords:** ACADL, adipocyte, differentiation, mechanism, RNA-seq, TNF signaling pathway

## Abstract

**Simple Summary:**

*ACADL* long-chain acyl-CoA dehydrogenase, not only a key enzyme in the initiation step of fatty acid oxidation, but also plays an important role in adipocyte differentiation. In our study, we found that *ACADL* promotes intramuscular adipocyte differentiation in goats via the TNF signaling pathway. This study provides a theoretical basis for the study of adipocytes differentiation in goat intramuscular adipocytes.

**Abstract:**

Intramuscular fat (IMF) deposits help improve meat quality such as marbling, juicy, flavor and tenderness. Long-chain acyl-CoA dehydrogenase (*ACADL*) is a key enzyme for catalyzing fatty acid oxidation, and studies have shown *ACADL* is involved in the deposition and differentiation of intramuscular adipocytes. However, the effect of *ACADL* on intramuscular adipocytes differentiation in goats needs further study. In this study, to explore the mechanism of *ACADL* on the development of goat intramuscular adipocytes, we constructed an over-expression plasmids and a SI-RNA of *ACADL* to explore the function of *ACADL* on the development of goat IMF. It was found that overexpression of *ACADL* promoted the differentiation of goat intramuscular adipocytes, and promoted the expression of fat cell differentiation marker genes lipoprotein lipase *(LPL*), peroxisome proliferator activated receptor gamma (*PPARγ*), APETALA-2-like transcription factor gene (*AP2*), CCAT enhancer binding protein (*CEBPα*), preadipocyte Factor 1 (*Pref-1*) and CCAT enhancer binding protein (*CEBPβ*), and the opposite trend occurred after interference. In addition, we screened of this related tumor necrosis factor (TNF) signaling pathway by RNA-Seq. So, we validate the signaling pathway with inhibitor of TNF signaling pathway. In summary, these results indicate that *ACADL* promotes intramuscular adipocytes differentiation through activation TNF signaling pathway. This study provides an important basis for the mechanism of IMF development.

## 1. Introduction

Lamb has low cholesterol and excellent tenderness, so people pay great attention to the traits of mutton, including meat’s color, marbling, tenderness, and flavor. Adipose tissue plays an important role in maintaining energy homeostasis, supporting vital activity of the body [1]. Among the four important fat deposits (visceral, subcutaneous, intermuscular, and intramuscular fat (IMF)), IMF is considered one of the most important factors affecting meat quality, and its deposition helps to improve the quality of meat, such as marbling, juiciness, flavor, and tenderness [2,3]. Therefore, methods to improve of IMF deposition are important for the improvement of meat quality and studying the mechanism of adipogenesis [4]. Studies has shown that the adipogenesis process was associated with transcriptional factors, such as CCAT enhancer binding protein (CEBP): *CEBPβ*; *CEBPα*; *PPARγ,* and *SREBP-1* [5,6,7].

*ACADL* is the key enzyme in catalyzing fatty acid oxidation [8], which belongs to a family of four closely related, chain length-specific acyl-CoA dehydrogenases: very long chain (*ACADCL*), long chain (*ACADL*), middle chain (ACADM), and short chain acyl-CoA dehydrogenases (*ACADS*) [9]. Stylissatin A (SA) and its derivatives might suppress the β-oxidation of fatty acids by ACADL, and the accumulation of fatty acids on macrophages would inhibit the nuclear factor-kappa B (NF-κB) signaling pathway [10]. Furthermore, *ACADL* has been shown to be associated with the deposition and differentiation of yak IMF (Wang et al., 2020), and *ACADL* deficiency can lead to severe hepatic and cardiac lipidosis, hypoglycemia and hepatic insulin resistance [11,12]. However, the effect of *ACADL* on intramuscular adipocytes differentiation in goats needs further study.

In this study, the regulatory effect of *ACADL* on intramuscular adipocyte differentiation in goats was studied by overexpression and interference. It was found that overexpression of *ACADL* promoted the differentiation of goat intramuscular adipocytes, and promoted the expression adipocytes differentiation marker genes *LPL*, *PPARγ*, *AP2*, *CEBPα*, Pref-1, and *CEBPβ*, and the opposite trend occurred after interference. To further explore the mechanism, RNA-seq techniques were performed to compare the differences between overexpression *ACADL* and control groups. We selected the TNF-signaling pathway from RNA-Seq results based on KEGG enrichment analysis and *ACADL* association pathway research reports. We investigated whether ACADL promotes intramuscular fat cell differentiation through the TNF-signaling pathway. Our experiments provide sufficient evidence that overexpression of *ACADL* promotes intramuscular preadipocytes differentiation in goats by activating the TNF-signaling pathway, and that the promoted trend is restored after the action of TNF-signaling pathway inhibitors (S1029). The above results show that the specific mechanism of *ACADL* to promote intramuscular adipocyte differentiation is elaborated.

## 2. Materials and Methods

### 2.1. Cell Culture

Our experiments meet the requirements of ethical treatment of Experimental Animals of China, and meet the requirements of the “List of Ethical Treatment of Laboratory Animals in China”. We purchased 1-year old Jian Zhou Da-er goat from Dageda animal husbandry (Sichuan, China). Isolation of intramuscular pre-adipocytes and culture of goat pre-adipocytes were both reported previously [13,14]. Briefly, the intramuscular adipocytes were isolation by using Type I collagenase. The longissimus dorsi were excised from 7-day-old goats and minced. Collagenase Type Ⅱ (37 °C, 1.5 h) (Sigma, Shanghai, China) and collagenase Type Ⅰ (37 °C, 1 h) (Sigma) are used to digest the pellets of longissimus dorsi. Enzymatic digestion was terminated by the same volume of DMEM/F12 (Hyclone, Logan, UT, USA) supplemented with 10% FBS (Gemin, Beijing, China). The suspension was filtered through a 75 µm nylon cell strainer, and then centrifuged at 750 rpm/min for 5 min. After disposing of the red blood cell lysed solution, the suspension was centrifuged at 750 rpm/min for 5 min again. The preadipocytes are resuspended in DMEM/F12 supplemented with 10% FBS, and diluted to a final concentration of 106 cells/ml. The cells were cultured at 37 °C under a humidified atmosphere containing 5% CO_2_.

### 2.2. Construction of Overexpression Vectors and siRNA of ACADL

In the early stage, we successfully cloned the goat *ACADL* gene using the liver tissue of goats as a template (stored by key laboratory of Qinghai-Tibetan Plateau Animal Genetic Resource Reservation and Utilization, Ministry of Education, Southwest Minzu University). In this study, we successfully built the *ACADL* overexpression vector (OE: PCDNA3.1-ACADL) with the control group PCDNA3.1. The goat *ACADL* interference sequence (*n* = 3) and negative control (NC) sequence were synthesized by Shanghai Gene pharma Bio Company. The information of *ACADL* shown in Table 1.

### 2.3. Oil Red O and Bodipy Staining

The staining method was performed by Xiong et al. [15,16]. Briefly, fix cells with 4% formaldehyde for 30 min (500 μL per well) and washed three times with PBS (800 μL per well, reaction for 5 min). Cells were then stained for 20 min using oil red O solution (Solaibo, Beijing, China) and Bodipy solution. After staining, the cells were imaged using a microscope (Olympus TH4, Tokyo, Japan) to obtain a magnified view. Finally, the oil red dye was extracted with 100% isopropanol and the extent of differentiation was determined by measuring the absorbance at 490 nm.

### 2.4. Cell Transfection

When goat intramuscular preadipocytes growth was confluent to 80%, we transfected ACADL overexpression and siRNA into cells according to the tubrofect transfection reagent (Thermo scientific, Waltham, MA, USA) instructions. We aspirated the medium 4 h before transfection, re-added 2 mL of fetal bovine serum medium (Hyclone, Logan, UT, USA), incubated for four hours, and transfected 4 μL of transfection reagent (Thermo, Waltham, MA, USA), 400 μL of OPTI-MEM (Gibco, Shanghai, China), and 1000 ng of no-load PCDNA3.1 or ACADL overexpression plasmid (and corresponding plasmid). Adipocytes were collected 48 h after induction of differentiation.

### 2.5. qRT-PCR

Total RNA was extracted by Trizol reagent and reversely transcribed into cDNA. The effect of overexpression *ACADL* and silencing *ACADL*, and other marker genes (*SREBP-1, LPL, PPARγ, AP2, CEBPα, PREF-1,* and *CEBPβ*) for differentiation through qPCR protocol. Ubiquitously expressed transcript gene (*UXT*) was used as internal housekeeping gene to normalize the mRNA levels. The PCR reaction procedure, including pre-degeneration (95 °C, 3 min), degeneration (95 °C, 10 s), annealing (60 °C, 10 s), and extension (72 °C, 15 s), 38 cycles for degeneration, annealing, and extension. The data were analyzed by the 2^−ΔΔCt^ method [17]. The information of marker genes primers, as shown in Table 1.

### 2.6. Library Preparation for Transcriptome Sequencing

Total RNA was used as input material for the RNA sample preparations. Briefly, mRNA was purified from total RNA using poly-T oligo-attached magnetic beads. Fragmentation was carried out using divalent cations under elevated temperature in First Strand Synthesis Reaction Buffer (5X). First strand cDNA was synthesized using a random hexamer primer and M-MuLV Reverse Transcriptase (RNase H). Second strand cDNA synthesis was subsequently performed using DNA Polymerase I and RNase H. Remaining overhangs were converted into blunt ends via exonuclease/polymerase activities. After adenylation of 3′ ends of DNA fragments, adaptor with hairpin loop structure were ligated to prepare for hybridization. In order to select cDNA fragments of preferentially 370~420 bp in length, the library fragments were purified with AMPure XP system (Beckman Coulter, Beverly, MA, USA). Then, PCR was performed with Phusion High-Fidelity DNA polymerase, Universal PCR primers and Index (X) Primer. At last, PCR products were purified (AMPure XP system) and library quality was assessed on the Agilent Bioanalyzer 2100 system [18].

### 2.7. Quality Control, GO and KEGG Enrichment Analysis of Differentially Expressed Genes

Raw data of fast format were firstly processed through in-house per scripts. In this step, clean data were obtained by removing reads containing adapter, reads 1 containing ploy-N and low-quality reads from raw data. At the same time, Q20, Q30, and GC content clean data were calculated. All the downstream analyses were based on the clean data with high quality.

Gene Ontology (GO) enrichment analysis of differentially expressed genes was implemented by the cluster Profiler R package, in which gene length bias was corrected. GO terms with corrected *p*-value less than 0.05 were considered significantly enriched by differential expressed genes. KEGG is a database resource for understanding high-level functions and utilities of the biological system, such as the cell, the organism, and the ecosystem, from molecular-level information, especially large-scale molecular datasets generated by genome sequencing and other high-through put experimental technologies http://www.genome.jp/kegg/ (accessed on 8 August 2021). We used cluster Profiler R package to test the statistical enrichment of differential expression genes in KEGG pathways.

### 2.8. Statistical Analysis

Take advantage graph-pad 5.0 to prepare graph. Variation was considered statistically significant at “*” *p* < 0.05, “**” *p* < 0.01. In this study, all experiments were repeated three times.

### 2.9. MTT Assay

MTT assays were used to detect the effects of TNF pathway inhibitors on cell viability. Goat intramuscular adipocytes were seeded in 96-well plates at a density of 2 × 10^3^ cells per well. After 0 and 12 h of cell treatment, we added 10 μL MTT reagent (Slarbio, Beijing, China) to each well, away from light, and then incubated the cells with 5% CO_2_ for 4 h at 37 °C. Finally, absorbance was measured at a wavelength of 490 nm.

## 3. Results

### 3.1. The effect of ACADL on the Intramuscular Preadipocytes Differentiation

As shown in Appendix A, the mRNA expression level of the *ACADL* gene was accompanied by a trend of increasing and then decreasing along with the differentiation of intramuscular adipocytes. Suggests that *ACADL* may be involved in the process of intramuscular fat fraction. To explore the effect of *ACADL* in regulating the differentiation of goat intramuscular adipocytes, we transfected *ACADL* over-expression vector (OE) and *ACADL*-interference (SI) and their corresponding to control vector and negative control (NC) groups into intramuscular adipocytes. The expression level of *ACADL* was largely upregulated compared to the vector (Figure 1A). For morphology, there were more lipid droplets stained by Bodipy dye and Oil red in the OE group than vector group (Figure 1B). Consistently, OD values at 490 nm in OE group was significantly higher than vector group (Figure 1C). Adipocytes differentiation was regulated by many adipogenesis transcription factors [19]. So, we detected the mRNA levels of these genes. The results showed that the overexpression of *ACADL* significantly upregulated the mRNA levels of *LPL (p* < 0.01), *PPARγ* (*p* < 0.01), *CEBPα (p* < 0.05), and PREF-1(*p* < 0.05). However, the mRNA levels of *AP2* (*p* < 0.05), and *CEBPβ* (*p* < 0.05) were downregulated (Figure 1D).

To further explore the effect of *ACADL* on intramuscular adipocytes, q-PCR was used to detect the expression of *ACADL* after *ACADL* interference. As shown in Figure 2A, the expression of *ACADL* decreased about 80% to that of NC group. Bodipy staining showed that the number of adipocytes decreased significantly after silencing *ACADL* (Figure 2B). Oil red O staining showed that the silence of *ACADL* inhibited lipid droplet accumulation. Consistently, OD value at 490 nm exhibited a decrease signal (Figure 2C). Conversely, the results showed silence of *ACADL* downregulated the expression levels of *LPL* (*p* < 0.05), *PPARγ* (*p* < 0.05), *CEBPα* (*p* < 0.05), and *PREF-1* (*p* < 0.05), while upregulated the expression of *SREBP-1* (*p* < 0.01) and *CEBPβ* (*p* < 0.01, Figure 2D). These data indicate that ACADL promotes the differentiation of goat intramuscular adipocytes.

### 3.2. RNA-Seq of ACADL Overexpression

After adipocytes differentiation for 48 h a significant increase in the degree of aggregation of lipid droplets (Figure 3A,B) and upregulation of the relative expression levels of *CEBPα*, *PPARγ,* and *LPL* marker genes can be observed (Figure 3C), which indicated that the differential fat cell model was successfully constructed.

### 3.3. Sequencing Data Quality Assessment

To illustrate the differentiation mechanism of *ACADL* on regulating goat differentiation intramuscular adipocytes differentiation more clearly, we performed RNA-Sequencing of *ACADL* overexpression (Appendix A). After quality filtration, 6 samples were used in this study regulatory effect of *ACADL* on differentiation of goat intramuscular adipocytes. The total RNA, RIN values, A260 nm/A230 nm, A260 nm/280 nm, and 28 s/18 s of 6 samples used in this study (control group NC, CS NC1 to 3, experimental group OE-ACADL, CS S1 to 3) met the criteria required for transcriptome sequencing (Appendix A, Table 2). In addition, in order to ensure the quality and reliability of data analysis, the original data need to be filtered (Appendix A). The results showed 14.26 G raw data and the effective data volume of each sample was distributed 6.35 G–6.61 G, average GC content was about 50.755 (Appendix A, Table 2), and the Q30 base distribution was 90.88–92.32, indicating that base reading of the sequencing data was obtained with high accuracy. The total mapping is above 94%, and the unique mapped reads were between 88.09% and 88.21%. In addition, transcripts were compared to genomic justice chains and antisense chains in similar proportions (Appendix A). The above indicates that the sequencing data are good and the data utilization rate is high, which provides a reliable basis for further analysis.

### 3.4. Analysis of Gene Expression

Quantitative analysis of the gene expression level was performed separately on each sequenced sample, and the amount of gene expression after overexpression was higher than that of the control group (Figure 4A). Correlation analysis of RNA-Seq samples showed that the samples were divided into two groups, and the relative relationship between the samples was 0.969 (Figure 4B). The co-expression Venn plot shows the number of genes co-expressed in the two groups/samples (Figure 4C), indicating that the expression patterns between samples in the group had a high degree of similarity. To identify differentially expressed genes (DEGs) between differential expressed mRNAs in intramuscular adipocytes after expression of *ACADL* were analyzed by DEG-Seq software ((*p* < 0.05), log_2_FC > 1), a total of 246 genes, of which 136 genes were upregulated and 110 genes were downregulated (Figure 4D). Then, clustering of differentially expressed mRNAs was performed, and intramuscular adipocytes sample before and after overexpression were clustered together (Figure 4E). It was worth noting that, from the statistical analysis of single nucleotide polymorphism (SNP), after overexpression of *ACADL,* the missense mutation in the mutation function of SNP (Appendix A), the INTRON in the mutant region (Appendix A), and the MODIFIER in the mutation effect were significantly better than in the control group (Appendix A). Together, *ACADL* overexpression causes more gene expression and leads to functional changes.

In the GO database, genes were divided into three catalogs: molecular function (MF), cellular component (CC) and biological process (BP), the number of gene-rich entries was metal-ion homeostasis (CC, 10), signaling receptor binding (MF,10), molecular function regulatory (MF, 10), receptor ligand activity (MF,6) and ion channel activity (MF,6) (Figure 5A). So, molecular function contains the most genes. To further confirm the potential function of DEGs in intramuscular adipocytes differentiation effect by *ACADL*. KEGG enrichment analysis showed that differentially expressed genes were significantly enriched in the top 20 signaling pathways (Figure 5B). Genes are significantly enriched in TNF, cytokine–cytokine receptor interactions, necrotizing apoptosis, tyrosine metabolism, and other signaling pathways (*p* < 0.05, Figure 5B). TNF pathway was the most likely and has rarely been studied in studies related to adipocytes differentiation. We used q-PCR analysis to confirm the accuracy of sequencing data, the same trend between the RNA-Seq and q-PCR was observed (Figure 5C–F).

### 3.5. TNF Pathway Rescues the Effect of ACADL Overexpression in Goat Intramuscular Adipocytes Differentiation

The RNA-Seq data showed that DEGs were enriched in the TNF pathway, and TNFAIP3 is one of main component to this pathway [20]. Reported research shows that lenalidomide S1093 (CC-5013) was a specific inhibitor in TNF signaling pathway [21], it was used to determine whether the inhibition TNF pathway rescues the effect of *ACADL* overexpression in goat intramuscular adipocytes differentiation. Since the concentration of the inhibitor has an effect on cell activity, as well as adsorption capacity [22]. To ensure the accuracy of the experiment and to reduce other interfering factors through the possible reduction in the experiment, we set the effect of different concentrations (0.1 μM, 0.3 μM, 0.5 μM, 1.5 μM, and 3.5 μM) on the *ACADL* expression for the detection of inhibitor efficiency, and the results showed that inhibitor efficiency was optimal at 2.5 μM (Figure 6A). MTT results showed that there was no difference of OD values in each group when compared to the blank control group at the same time point (Figure 6B), which indicates that cell viability is not affected. TNF inhibitor treat had fewer number of adipocytes lipid droplets (Figure 6C). In accordance with the above results, the inhibitors rescue the expression level of *ACADL* after overexpression (Figure 6C). OD value results trend was consistent with Figure 6C (Figure 6D). The OD value of the inhibitor treatment group was significantly lower than that of the control group, and had an attenuating effect on the *ACADL* overexpression treatment group.

Moreover, under the action of inhibitors, the TNF-signaling pathway gene appears to have an upregulated trend (Figure 7a), *ACADL* and adipocytes-related genes expression appears to be reversed or weakened in response to the overexpression in *ACADL* (Figure 7b–j).

## 4. Discussion

ACADL has an important role in intramuscular adipocytes differentiation. Research shows that *ACADL* deficiency causes cardiac lipids and hypoglycemia [12]. Long-chain acyl-CoA dehydrogenase deficiency causes mitochondrial dysfunction and leads to hepatic steatories and hepatic insulin resistance [23]. Research on the effects of *ACADL* on fat has mainly focused on livestock, such as yaks and pigs, however, research on goats needs to go further. In this study, we investigated the role of *ACADL* regulating on the goat intramuscular adipocytes by overexpression *ACADL* and silencing *ACADL*. Increased aggregation of lipid droplets in intramuscular adipocytes after overexpression of *ACADL* was observed by Oil red and Bodipy (fluorescent) staining, but decreased the degree of aggregation of lipid droplets in IMF adipocytes after interference. As is known to all, adipocytes differentiation determined by hyperplasia and hypertrophy of intramuscular adipocytes [24]. Intramuscular adipocytes differentiation is a complex event involving many biological processes and transcription factors that are responsible for inducing mature adipocyte formation [25]. *PPARγ*, *CEBPβ*, and *CEBPα* are considered as the important regulators of adipocytes [26]. *AP2*, *SREBP-1,* and *LPL* have been considered positive regulator for adipocyte differentiation, while *PREF-1* have been identified as factors negatively regulators for this [27]. In this study, after the overexpression of *ACADL*, the expression of *LPL*, *PPARγ*, *CEBPα,* and *PREF-1* were enhanced. In general agreement with the expected results, the expression levels of *SREBP-1*, *AP2,* and *CEBPβ* were downregulated after *ACADL* silencing. These results suggest that *ACADL* may promote intramuscular adipocyte differentiation through upregulation of *LPL*, *PPARγ*, and *CEBPα*. In IMF adipocytes, whether *ACADL* directly or indirectly modulates the level of the adipogenesis transcription gene mRNA directly or through molecules requires further exploration.

To reveal the potential specific molecular mechanism, RNA-sequencing was performed. Quantitative analysis showed that the number of overexpressed genes was higher than that of the control group (VECTOR). There are 246 differential expression genes (DEGs) intramuscular adipocytes after *ADADL* overexpression treatment, and the upper 136 genes are more than 110 down-regulated genes. *ACADL* overexpression causes more gene expression and leads to functional changes. We select the TNF-signaling pathway from KEGG enrichment analysis. Interestingly, studies have shown that TNF promotes the differentiation of intramuscular adipocytes by activating the downstream pathways by secreting the TNFα factor [28]. Similarly, TNF mediates activation of the transcription factor NFκB, which, in turn, regulates the differentiation and maturation of adipocytes [29,30]. So, we thought the TNF-signaling pathway is the best choice among the 20 pathways for KEGG analysis of DEGs. Sequencing yielded four differentially expressed genes: Tumor necrosis factorα inducer protein 3 *(TNFAIP3*), E-selectin (*SELE*), CC-chemokine 5 (CCL5), and CC-chemokine 20 (*CCL20*) enriched in this pathway. *TNFAIP3* is the main component of the TNF-signaling pathway, *TNFAIP3* is the key factor for mediating the secretion of TNFa [31]. *TNFAIP3* inhibits the secretion of TNFa in the TNF-signaling pathway, which, in turn, affects the downstream pathway of the TNF signaling pathway to regulate adipocyte differentiation [32,33]. E-selectin is a key adhesion molecule, and research has reported that elevated soluble E-selectin levels have been identified in hypertension and diabetes [34]. In addition, E-selectin expression is stimulated by exposure to tumor necrosis factor-alpha [35]. *CCL5* and *CCL20* all belong to the chemokines, and play roles in various pathologic conditions [36]. TNFα directly affects a number of chemokines, including *CCL20* [37], and *CCL5* selectively activates the downstream JNK pathway of the TNF pathway [38,39,40]. We verified their expression levels through qPCR analysis. Consistently, overexpression of *ACADL* downregulated expression of *TNFAIP3*, *SELE*, *CCL5*, and *CCL20*.

To determine the role of *ACADL* in promoting adipocytes differentiation in goats through the TNF-signaling pathway, we added TNF-pathway inhibitors (S1029) to intramuscular adipocytes for further validation. In intramuscular adipocytes, *ACADL* overexpression activates TNF pathway, which subsequently lead to increased differentiation of intramuscular adipocytes. Interestingly, the TNF inhibitor treatment OE group rescued the phenotype caused by *ACADL* expression in a specificity dose manner. Preadipocytes are characterized by the number of cells and enlargement. The TNF-signaling pathway plays an important role in both [41]. In the inhibitor treatment group, the expression of *TNPAIP3*, *SELE*, *CCL5*, and *CCL20* was upregulated, which was contrary to the sequencing data and ACADL overexpression treatment group, and demonstrated that the inhibitor is a potent inhibitor of the TNF pathway. The apparent phenomena showed that the inhibitor treatment group significantly reduced the degree of cell differentiation and lipid droplet aggregation, which had a weakening effect on the *ACADL* overexpression group. Expression of *TNFAIP3*, *SELE*, *CCL5,* and *CCL20* in the inhibitor-treated group showed an upward trend, the fat adipocyte differentiation-related genes also showed a reverse trend, and the expression level of S019 in the OE treatment group showed a weakening trend. Therefore, we speculate that *ACADL* regulates the TNF-signaling pathway and subsequently affects pre-adipocyte differentiation by altering cellular lipid droplet aggregation

## 5. Conclusions

The overexpression of *ACADL* promotes adipocytes differentiation and silencing *ACADL* decreased adipocytes differentiation. Further study showed that TNF-signaling pathway is involved in the regulation of *ACADL* on the differentiation of adipocytes. In summary, *ACADL* promotes goat adipocytes differentiation by activating the TNF-signaling pathway.

## Figures and Tables

**Figure 1 animals-13-00281-f001:**
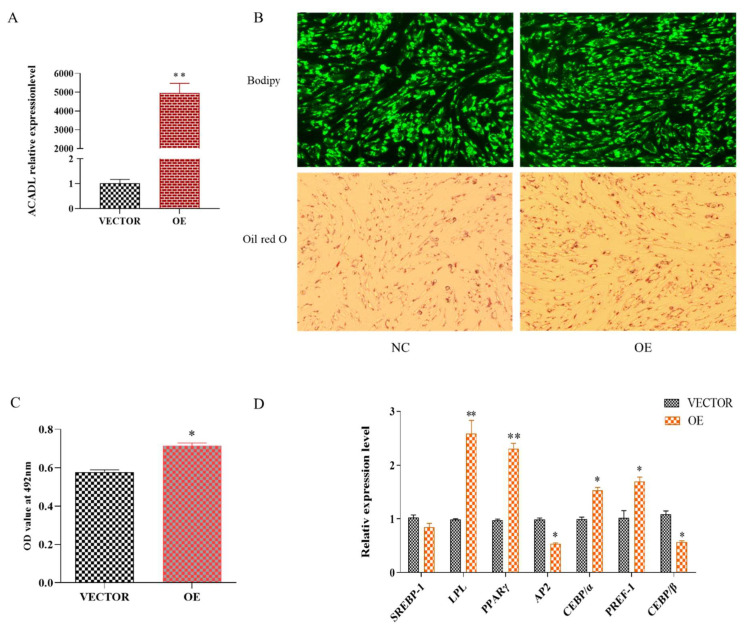
Over-expression of goat *ACADL* promotes intramuscular adipocytes differentiation. (**A**) mRNA expression of *ACADL* was detected by qPCR after over-expressing *ACADL* for 48 h intramuscular adipocyte differentiation. (**B**) Photos of Bodipy staining and Oil red staining of the cells in the tests group (OE) and negative control group (NC) during intramuscular adipocyte differentiation. (**C**) The OD value of Oil red O staining at 492 nm during intramuscular adipocytes differentiation. (**D**) Effect of goat ACADL overexpression on the expression of adipocytes related genes. “*” significant difference and “**” very significant difference.

**Figure 2 animals-13-00281-f002:**
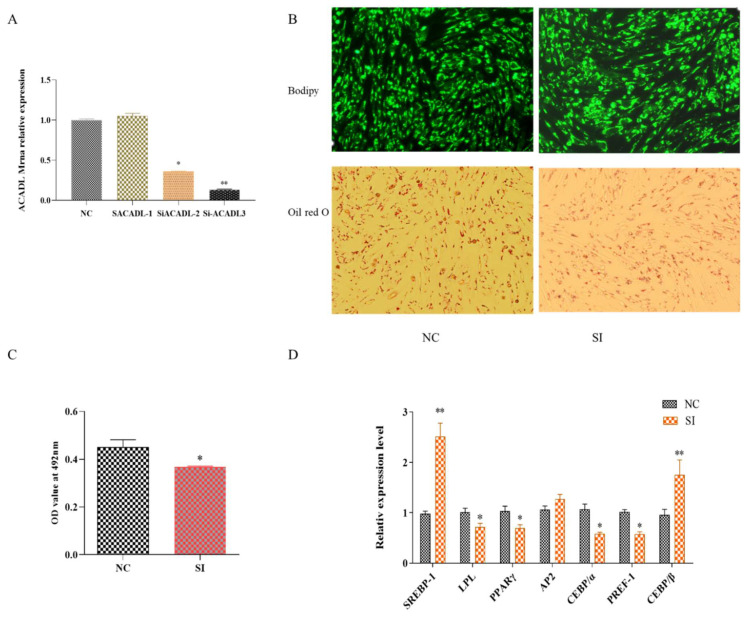
The effect of *ACADL* interference on goat adipocyte differentiation. (**A**) Expression efficiency detection after *ACADL* interference. (**B**) Morphology observation of Oil red O staining. (**C**) Oil red O staining showed the OD value at 492 nm during the differentiation of intramuscular adipocytes. (**D**) Effect of goat *ACADL* overexpression on adipocytes related gene’s expression. “*” significant difference and “**” very significant difference.

**Figure 3 animals-13-00281-f003:**
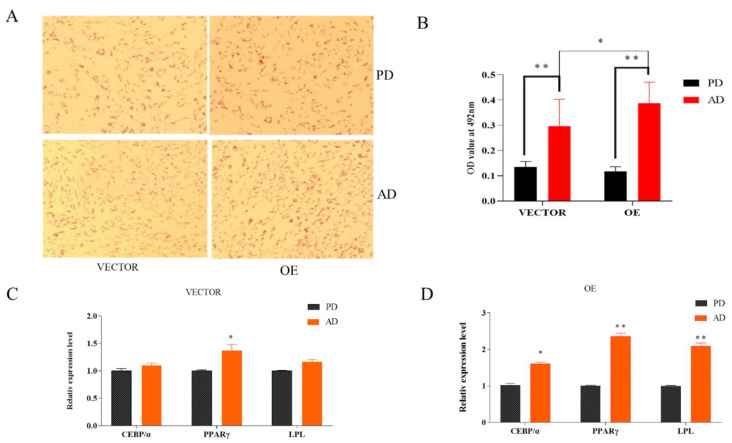
Adipocytes differentiated for 2 days. (**A**) Oil red staining, pre-differentiation (PD), after differentiation (AD). (**B**) OD value of preadipocytes and adipocytes, relative expression of *ACADL*. (**C**,**D**) Expression level of marker genes. “*” significant difference and “**” very significant difference.

**Figure 4 animals-13-00281-f004:**
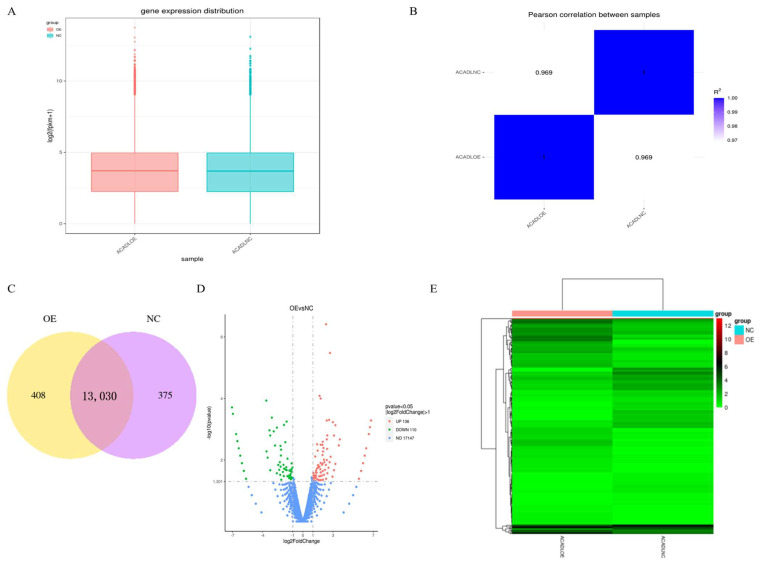
Gene expression analysis. (**A**) Distribution of gene expression. (**B**) Heat map of sample-to-sample correlation. The horizontal ordinate coordinates in the plot are the squares of the correlation coefficients for each sample. (**C**) Co-expression of Venn diagram. The number of genes uniquely expressed in each group, overlapping regions showing the number of genes co-expressed in two groups. (**D**) Differential gene volcano. The abscissa is the log_2_FoldChange value, the ordinate is log_10_padjust or -log_10_p-value, and the blue dotted line indicates the threshold line of the differential gene-screening criteria. (**E**) Heat map of differentially expressed gene clusters. Abscissa is the sample name, and the ordinate coordinate was the value of the FFPKM normalization of the differential gene, the redder the color, the higher the expression, the greener, and the lower the expression.

**Figure 5 animals-13-00281-f005:**
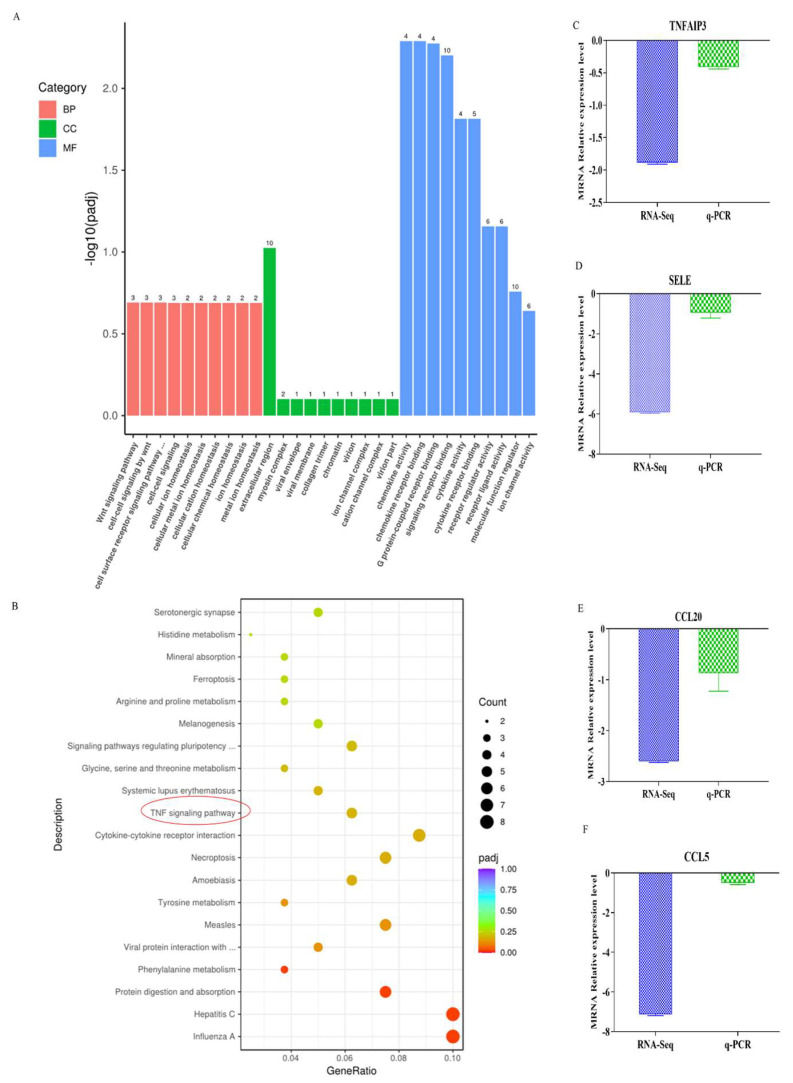
Enrichment analysis. (**A**) GO enrichment analysis histogram. The abscissa was GO term, the ordinate was the significance level of GO term enrichment, represented by −log_10_ (*p*-value, and different colors indicate different functional classifications. (**B**) KEGG enrichment scatter plot. The abscissa coordinate is the ratio of the number of differential genes to the total number of differential genes annotated to the KEGG pathway, and the ordinate is the KEGG pathway, red circles indicate the selected pathway (**C**–**F**) Validation of representative by q-PCR (*n* = 5) Log_2_*p*-value. TNFAIP3: tumor necrosis factor induced protein 3, *CCL5*: chemokine ligand 5, *CCL20*: chemokine ligand 20, *SELE*: selectin endothelial an adhesion molecule.

**Figure 6 animals-13-00281-f006:**
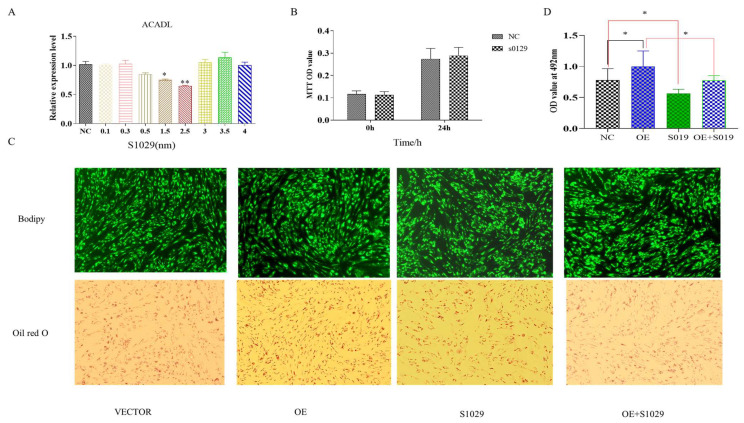
Overexpression of *ACADL* promotes intramuscular adipocytes differentiation through TNF-signaling pathway. (**A**) The mRNA level of *ACADL* in inhibitors of different concentrations. (**B**) MTT for the detection of cell activity over different time period. Screening of inhibitors for optimal concentrations (Unit nm). (**C**) The image of Bodipy and Oil red O staining of intramuscular adipocytes in VECTOR, OE, inhibitor, and OE combined with inhibitor. (**D**) OD value of VECTOR, OE, inhibitor, and OE combined with inhibitor. “*” significant difference and “**” very significant difference.

**Figure 7 animals-13-00281-f007:**
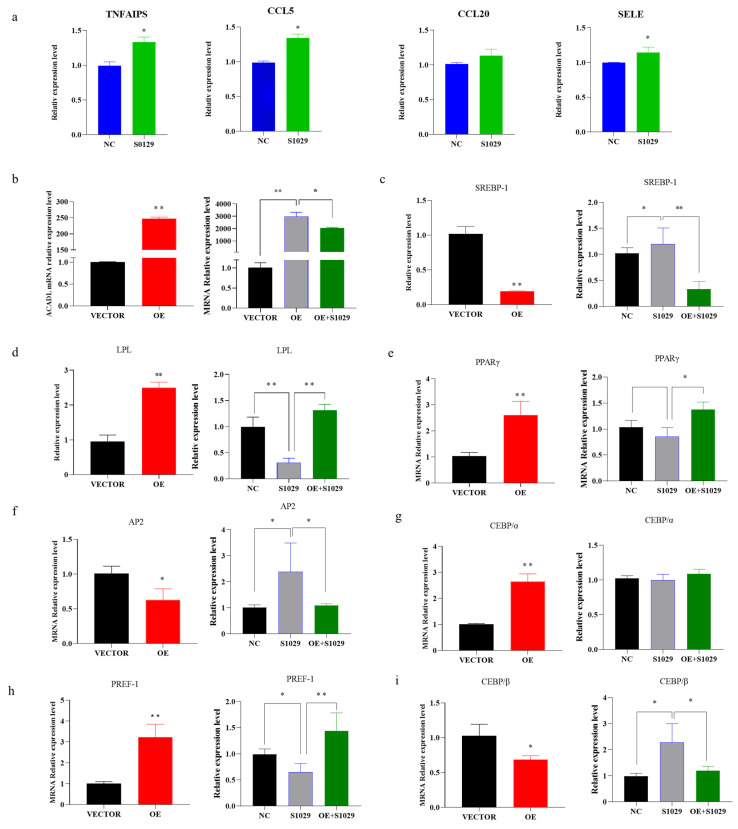
The recovery of gene expression is under the action of inhibitors. (**a**) Expression of TNF signaling pathway genes after the action of the inhibitor. (**b**) The expression of *ACADL* in different treatment control groups (OE, s1029, OE + S1029). (**c**–**i**) The expression level of differentiation marker genes in different treatment control groups. The mRNA level of *ACADL* (**c**), *SREBP-1* (**d**), *LPL* (**e**), *PPARγ* (**f**), *AP2* (**g**), *CEBPα* (**h**), *PREF-1* (**i**), and CEBPβ in VECTOR, OE, and OE combined with inhibitors. n = 6, “*” *p* < 0.05, “**” *p* < 0.01.

**Table 1 animals-13-00281-t001:** Information of sequences and primers.

Sequence Name	Sequence	Tm/°C
Si-ACADL-1	GCCUGUACAAUUUGAAUAUTT AUAUUCAAAUUGUACAGGCTT	
Si-ACADL-2	CCACCCAUUAGUGACAAAUTT AUUUGUCACUAAUGGGUGGTT	
Si-ACADL-3	GCUCUUGCAUGAGGUAAUATT UAUUACCUCAUGCAAGAGCTT	
*SREBP-1*	S: AACATCTGTTGGAGCGAGCA A: TCCAGCCATATCCGAACAGC	60 °C
*LPL*	S: GAGGCCTTGGAGATGTGGAC A: AATTGCACCGGTACGCCTTA	60 °C
*PPARγ*	S: AAGCGTCAGGGTTCCACTATG A: GAACCTGATGGCGTTATGAGAC	60 °C
*AP2*	S: TGAAGTCACTCCAGATGACAG A: TGACACATTCCAGCACCAG	58 °C
*CEBPα*	S: CTCCGGATCTCAAGACTGCC A: CCCCTCATCTTAGACGCACC	60 °C
*PREF-1*	S: CCTGAAAATGGATTCTGCGACG A: GACACAGGAGCACTCGTACTG	60 °C
*CEBPβ*	S: CAACCTGGAGACGCAGCACAAG A: GCTTGAACAAGTTCCGCAGGGT	60 °C
*UXT*	S: GCAAGTGGATTTGGGCTGTAAC A: ATGGAGTCCTTGGTGAGGTTGT	60 °C

**Table 2 animals-13-00281-t002:** Raw data and quality control data of each sample group.

Sample	Raw Reads	Raw Bases	Clean Reads	Clean Bases	Error-Rate	Q20	Q30	GC-Pct
ACADLOE	48,014,900	7.2 G	44,044,956	6.61 G	0.03	97.03	92.32	51.91
ACADLNC	46,690,324	7 G	42,324,680	6.35 G	0.03	96.51	90.88	50.6

## Data Availability

The data used to support of this study are available from the corresponding author on reasonable request.

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
