# Peer review of "ACADL Promotes the Differentiation of Goat Intramuscular Adipocytes"

_animals, 2023, doi:10.3390/ani13020281_

Round 1
Reviewer 1 Report
1. English needs to be modified
Eg: In Abstract, change the value of contributions in line 17 to contributions to; The plastid in line 23 is changed to plastids; 29 lines: adipocytes is changed to adipocytes'; The isoform of Line 77 was changed to isoform, etc;
2. Format needs to be modified
Eg: The gene should be italicized; 69 lines (incomplete; μ Space between L and number
3. More details need to be added to the method
Eg: For each cell reagent and machine used, specific information such as model and company should be indicated
4. Whether TNF pathway is significantly enriched should be explained
Eg: The specific P value is given in the F5B diagram to explain whether there is significant enrichment.
Author Response
- English needs to be modified
Eg: In Abstract, change the value of contributions in line 17 to contributions to; The plastid in line 23 is changed to plastids; 29 lines: adipocytes is changed to adipocytes'; The isoform of Line 77 was changed to isoform, etc;
Response:We have modified these content according your suggestion.
- Format needs to be modified
Eg: The gene should be italicized; 69 lines (incomplete; μ Space between L and number)
Response:We have modified these content according your suggestion, We changed all genes to italics, and changed the space between μl and number.
- More details need to be added to the method
Eg: For each cell reagent and machine used, specific information such as model and company should be indicated.
Response The specific information of the reagent has been clearly marked in the text
- Whether TNF pathway is significantly enriched should be explained
Eg: The specific P value is given in the F5B diagram to explain whether there is significant enrichment.
Response We have modified the content according your suggestion. Line 308- 310. With significant differences in the first 20 signaling pathways. Genes are significantly enriched in TNF, cytokine-cytokine receptor interactions, necrotizing apoptosis, tyrosine metabolism and other signaling pathways ( P < 0.05, Figure 5B)
Reviewer 2 Report
The methods and data seem appropriate and the conclusions are supported. Unfortunately the manuscript is very difficult to understand and needs a thorough re-write to make it clear. For example,
Line 20 - "..ACADL is involved in the deposition and differentiation of intramuscular." - intramuscular what?
Line 81-82 - "..It has been reported that it inhibits the β-oxidation of fatty acids by ACADL,.." - what inhibits the β-oxidation of fatty acids?
Lines 99-100
Line 153 - What is the reference gene? If someone tried to repeat the study, that would important to know. Evidence that the reference gene is expressed at a consistent level?
There are errors of format. Line 25 - PPar should be PPAR. Line 121 - 106 or 10E6?
These are just some of the problems.
Although a list of abbreviations is given, the definitions are also given in the text.
Author Response
1.Line 20 - "..ACADL is involved in the deposition and differentiation of intramuscular." - intramuscular what?
Response We have correct the content. Line 19-20. ACADL is involved in the deposition and differentiation of intramuscular adipocytes.
2.Line 81-82 - "..It has been reported that it inhibits the β-oxidation of fatty acids by ACADL,.." - what inhibits the β-oxidation of fatty acids?
Response: This is a gap in our description, and we have corrected these problems.
Line 83-87 Stylissatin A (SA) and its derivatives might suppress the β-oxidation of fatty acids by ACADL, and the accumulation of fatty acids on macrophages would inhibit the nuclear factor-kappa B (NF-κB) signaling pathway
Lines 99-100
3.Line 153 - What is the reference gene? If someone tried to repeat the study, that would important to know. Evidence that the reference gene is expressed at a consistent level?
Response: In response to your question, This is our translation error that led to your misunderstanding. The internal reference gene, which is commonly translated as the housekeeping gene, we have corrected it.
Housekeeping genes: their expression in various tissues and cells is relatively constant, and they are often used as a reference substance when detecting the change of gene expression level, the function is to correct the amount of samples, and there are errors in the stage process experiment to ensure the accuracy of experimental results.
4.There are errors of format. Line 25 - PPar should be PPAR. Line 121 - 106 or 10E6?
Response:We have changed the content according your suggestion.
5.These are just some of the problems.
Although a list of abbreviations is given, the definitions are also given in the text.
Response: We've added definitions of aliases in the text where appropriate.
Reviewer 3 Report
1. Here are some editing errors, such as Line 135 AP2((APETALA2)
2. Abbreviations of 60-95 names require the creation of a three-line table?
3. Line 299-300 There is a controversy here, for ACADL overexpression is to promote intramuscular fat cell differentiation, and PREF-1 is a negative regulator of intramuscular fat cell differentiation, ACADL overexpression after its expression is upregulated, which is unreasonable.
4. Line 330-335 “What does Figure 3 mean? Is it a batch with the previous cell diagram? Why did you do this experiment, is it necessary?
5. Line 717 Is it necessary to supplement Figure S3? Does ACADL cause this change after overexpression in adipocytes?
6. Figure 6: Different sizes of cell plots? Please correct it
7. The supplementary figure in Figure 2 cannot be seen clearly, and the clarity is too low
Author Response
1.Here are some editing errors, such as Line 135 AP2((APETALA2)
Response: We've corrected the editing mistakes in the text.
2.Abbreviations of 60-95 names require the creation of a three-line table?
Response: We have modified the content according your suggestion.
3.Line 299-300 There is a controversy here, for ACADL overexpression is to promote intramuscular fat cell differentiation, and PREF-1 is a negative regulator of intramuscular fat cell differentiation, ACADL overexpression after its expression is upregulated, which is unreasonable.
Response: For the negative regulators of adipocyte differentiation, studies have shown that the expression of PREF-1 is also increased on the production of obesity, here due to the different mechanisms of ACADL promoting adipocyte differentiation, ACADL generally promotes adipocyte differentiation, but the regulatory mechanism of PREF-1 here is slightly different.
4.Line 330-335 “What does Figure 3 mean? Is it a batch with the previous cell diagram? Why did you do this experiment, is it necessary?
Response: Figure 3 is the preparation before sequencing, the first is to ensure that the adipocytes have induced differentiation; The second is guaranteed to ensure that the effects of ACADL overexpression on intramuscular fat cells are guaranteed. These are ultimately all to ensure the accuracy of the experiment.
5.Line 717 Is it necessary to supplement Figure S3? Does ACADL cause this change after overexpression in adipocytes?
Response: In this study, this result is of some importance, ACADL overexpression promotes intramuscular adipocyte differentiation, which may promote adipocyte differentiation by causing changes in the mononucleotide polymorphisms of certain genes.
6.Figure 6: Different sizes of cell plots? Please correct it
Response: We have modified the figure.
7.The supplementary figure in Figure 2 cannot be seen clearly, and the clarity is too low.
Response: We have modified the clarity of the figure
Round 2
Reviewer 2 Report
The manuscript appears much improved. Please correct minor errors - Lines 70, 91, 97; Incomplete sentences - Lines 75, 283; Line 175 - delete 'using enzyme labeled instrument'; Please adjust table 2 spacing, column 4.
Author Response
1. The manuscript appears much improved. Please correct minor errors - Lines 70, 91, 97;
Response: We have fixed these errors based on your suggestions.
2. Incomplete sentences - Lines 75, 283;
Response: We have modified the content according your suggestion.
3. Line 175 - delete 'using enzyme labeled instrument'; Please adjust table 2 spacing,column
Response: We have modified the content according your suggestion.